# Near-Infrared 810 nm Light Affects Porifera *Chondrosia reniformis* (Nardo, 1847) Regeneration: Molecular Implications and Evolutionary Considerations of Photobiomodulation–Animal Cell Interaction

**DOI:** 10.3390/ijms24010226

**Published:** 2022-12-23

**Authors:** Andrea Amaroli, Eleonora Tassara, Sara Ferrando, Stefano Aicardi, Claudio Pasquale, Marco Giovine, Marco Bertolino, Angelina Zekiy, Marina Pozzolini

**Affiliations:** 1Department of Orthopedic Dentistry, Faculty of Dentistry, First Moscow State Medical University (Sechenov University), 119991 Moscow, Russia; 2Department of Earth, Environment and Life Sciences (DISTAV), University of Genova, Via Pastore 3, 16132 Genova, Italy; 3Department of Surgical and Diagnostic Sciences, University of Genoa, 16132 Genoa, Italy

**Keywords:** low-level laser therapy, light therapy, Porifera, evolution, tissue regeneration, inflammation, collagen, heat shock protein, tumour necrosis factor, transforming growth factor, Wnt

## Abstract

Chemotrophic choice as a metabolic source of energy has characterised animal cell evolution. However, light interactions with animal cell photoacceptors that are able to increase energetic metabolism (photo-biomodulation (PBM)) have been previously described. In the present study, we cut three specimens of *Chondrosia reniformis* into four equal parts (12 fragments), and we irradiated the regenerating edge of six fragments with the previously characterised 810 nm near-infrared light, delivered at 1 W, 60 J/cm^2^, 1 W/cm^2^, and 60 J in a continuous-wave mode for 60 s through a flat-top hand-piece with a rounded spot-size area of 1 cm^2^. Six fragments were irradiated with 0 W for 60 s as the controls. We performed irradiation at the time 0 h and every 24 h for a total of five administrations. We monitored the regeneration process for five days (120 h) in aquaria by examining the macroscopic and histological changes. We analysed the gene expression profile of the inflammatory processes, apoptosis, heat stress, growth factors, and collagen production and determined oxidative stress enzyme activity and the total prokaryotic symbiont content. PBM sped up *C. reniformis* regeneration when compared to the controls. Particularly, transforming growth factor TGF3 and TGF6 upregulation during the early phase of regeneration and TGF5 upregulation 120 h postinjury in the irradiated samples supports the positive effect of PBM in sponge tissue recovery. Conversely, the expression of TGF4, a sponge fibroblast growth factor homologue, was not affected by irradiation, indicating that multiple, independent pathways regulate the TGF genes. The results are consistent with our previous data on a wide range of organisms and humans, suggesting that PBM interaction with primary and secondary cell targets has been conserved through the evolution of life forms.

## 1. Introduction

Visible (VIS) and near-infrared (NIR) light interaction with life forms has been documented extensively in plant cells. Indeed, plants use photons as a source of energy and the conversion of this physical energy into chemical energy has been documented through evolution in photosynthesising organisms [1]. Conversely, chemotrophic choice as a metabolic source of energy has characterised animal cell evolution. Therefore, even though all life forms need energy for survival, animals have limited light reception to specialised molecules (photoreceptors) capable of harnessing light to enable their vision and circadian rhythm or to affect the production of vitamin D [2].

In recent decades, researchers have described photo-biomodulation (PBM) or low-level laser therapy, as a nonthermal and nonionising light interaction with animal cells [3,4]. There are two types of molecules in an animal cell that could interact with photons: the photoreceptors specialised to absorb light, and some nonspecialised molecules–photoacceptors–that are able to be energised through the transfer of photonic energy from VIS and NIR wavelengths, including light-emitting diodes (LEDs), lasers, and broadband light [5]. Mitochondrial cytochromes, haem-containing proteins, nitrosyl-iron complexes, thiol groups, flavins, water, and lipids likely act as photoacceptors [6]. Consequently, the photo-manipulation of cellular metabolism may also be observed in nonplant cells [4,6].

Recently, Hamblin et al. [7] and Amaroli et al. [8] suggested the potential of life forms to respond to PBM. Despite PBM being considered a promising medical therapy and being extensively investigated in that field, the evolutionary interpretation in terms of conserving both light–organism interactions and cellular pathways has not been discussed, partly due to a lack of consistent data. Therefore, in previous studies, we have followed a standard experimental setup and have investigated the effect of 810 nm NIR light provided at 1 W of power, 60 J/cm^2^ of fluence, 1 W/cm^2^ of power density, and 60 J of energy, delivered in a continuous-wave (CW) mode for 60 s through a flat-top hand-piece (FT-HP) to a rounded spot-size area of 1 cm^2^ (hereinafter 810 nm-1W PBM) on a wide range of organisms, such as *Paramecium primaurelia* (Protozoa), [9,10,11,12,13] *Dictyostelium discoideum* (Protozoa), [14] *Dendrobaena veneta* (Annelida) [15,16] *Paracentrotus lividus* (Echinodermata), [17] *Branchiostoma lanceolatum* (Cephalochordate), [18] *Mus musculus* (Gnathostomata, Mammalia), [19,20] *Bos taurus* (Gnathostomata, Mammalia) [21], and *Homo sapiens* (Gnathostomata, Mammalia) [22,23,24,25]. We have described the effects on cellular/tissue homeostasis, energetic metabolism, cell proliferation, and differentiation, as well as illness recovery.

In order to ensure the consistency of our data and to deepen the discussion, we have investigated the effect of 810 nm-1 W PBM on Porifera. The members of phylum Porifera are considered the simplest and the oldest surviving metazoan lineages, and for this reason, they have a key role in the search for the origins of the various multicellular metazoan processes [26]. Indeed, Müller [27,28] described that in the origin of metazoan complexity, the transition from the colonial stage of organisation to the integrated stage occurred in the hypothetical ancestor Urmetazoa and its evolution into Porifera as the first integrated animals. Marine sponges have high regenerative power, and numerous studies have been conducted to elucidate the relevant molecular mechanisms [29,30,31]. Sponge regeneration seems to require the mesohyl of totipotent stem cells known as archaeocytes [32]. Researchers have emphasised that sponge wound healing occurs through ancient genes and regulatory networks that also underlie tissue processes in higher animals [33,34].

The experimental model we examined in this study is *Chondrosia reniformis*, a member of the Demospongiae class. It is widely distributed in the Mediterranean Sea on the shaded walls of the rocky bottom assemblages up to 50 m in depth [35]. Many of its characteristics–such as different physiological responses to changes in environmental conditions, [36] the incorporation of silica [37], and the production of collagen [38] and other secondary metabolites–have made this species interesting to the scientific community and for biotechnological purposes [38,39]. Moreover, the first nonspecific tumour necrosis factor (TNF)-mediated innate immune response was described in this sponge [40].

Therefore, to investigate the macroscopical and biomolecular effects of PBM on organisms, we performed a study on *C. reniformis* specimens exposed to the previously characterised 810 nm-1 W PBM. The predictor variable was the 810 nm-1 W PBM ability to affect cell metabolism in higher animals and humans [18,19,20,21,22,23,24,25,26]. The primary endpoint was to observe if 810 nm-1 W PBM can positively affect the regeneration process in an organism with a very simple body plan, considered an early animal ancestor.

The gene expression profile of the inflammatory processes, apoptosis, heat stress, growth factors, collagen production, oxidative stress enzyme activity, and total prokaryotic symbiont content were analysed for this purpose. The results were compared with our previous data in unicellular organisms, animals, and humans [9,10,11,12,13,14,15,16,17,18,19,20,21,22,23,24,25,26], and the molecular implication and evolutionary considerations on PBM–animal cell interaction are discussed.

## 2. Results

### 2.1. Sponge Morphology during Regeneration

During the first three days of regeneration, there were no significant morphological differences between the 810 nm-1 W PBM irradiated samples (Figure 1) and the control samples exposed to 810 nm-0 W PBM (Figure 1). However, after five days (120 h), the 810 nm-1 W PBM samples had a more rounded shape than the controls, especially Sponge 1. Here, a smaller cutting edge indicates more advanced regeneration. As shown in Appendix A, the average regeneration percentage of the three samples was ~70% in the sample irradiated versions and ~35% in the controls.

### 2.2. Histology and Archaeocyte Identification by Immunostaining

Masson’s trichrome-stained slides showed that 24 h after the injury, the regenerating edge of the control (Figure 2A) and PBM fragments (Figure 2B) was covered by spherulous cells and a few flattened cells that could resemble exopinacocytes. Musashi 1 (Msi-1) immunofluorescence (Figure 3A) allowed us to observe that, in the first hundreds of micrometres from the regenerating edge, archaeocytes were significantly abundant in the PBM fragments, although they were also present in control fragments (Figure 3B,C).

### 2.3. Gene Expression Profile

Heat shock proteins (HSPs) and nitric oxide synthase (NOS) could be modulated by both inflammatory and heat-stress factors [41,42,43]. Therefore, we evaluated the expression profiles of these genes in the cut sponges. After 24 h, HSP60 was significantly upregulated (30.47 ± 14.64-fold) in the 810 nm-1 W PBM samples compared with the control, while after 120 h, there were no significant differences compared with the 120 h controls (Figure 4A). Of note, during the regeneration, the control samples exhibited a significant increase in HSP60 mRNA (22.0 ± 8.89-fold) at 120 h postexcision, relative to 24 h postexcision. The HSP70 gene had the same expression pattern (Figure 4B), showing at (24 h postexcision) a significant upregulation (35.33 ± 14.64-fold); no significant differences were observed 120 h postexcision in the 810 nm-1 W PBM samples compared with their 120 h controls. However, at 120 h postexcision, in the nonirradiated samples, the significant upregulation (17.33 ± 5.51-fold) of the HSP70 gene was registered compared to the 24 h untreated control sample. When taken together, these results suggest that PBM treatment mainly leads to a faster response to HSP60 and HSP70 expression after tissue injury. Conversely, 24 h postexcision HSP90 gene expression was significantly downregulated in the PBM samples (0.45 ± 0.14 -fold) compared with their controls. The same trend was also observed 120 h postexcision, as HSP90 had significantly decreased by 3.30 ± 1.00-fold in the PBM samples relative to the control samples. These results indicate that PBM treatment in our model could act as a negative regulator of the HSP90 gene. As observed for the HSP60 and HSP70 genes, in the control samples, HSP90 expression increased by 3.15 ± 1.00-fold 120 h postexcision relative to 24 h postexcision (Figure 4C).

At 24 h and 120 h after cutting, there were no significant differences in NOS gene expression between the irradiated and nonirradiated samples (Figure 4D). However, it should be noted that from 24 to 120 h postexcision, NOS increased 3.45 ± 0.75-fold in the control samples and 1.33 ± 0.39-fold in the irradiated samples.

As shown in Figure 4E, in the PBM samples at 24 h postexcision, the TNF mRNA had increased 3.04 ± 0.72-fold compared with the controls, while at 120 h postexcision, a slight but significant TNF gene downregulation (0.75 ± 0.13) occurred when compared with the control, which suggests that the transcript levels return to the pre-PBM baseline. There were no differences between the 24 h and 120 h control samples.

The antiapoptotic protein Bcl-2 was upregulated significantly in the PBM samples at 24 and 120 h postexcision compared with the control samples (3.33 ± 0.36-fold and 1.11 ± 0.4-fold, respectively), indicating a reduction in apoptosis in the regenerating sponge tissue after irradiation, while no significant differences in Bcl-2 gene expression were observed between the 24 and 120 h control samples (Figure 4F).

In order to evaluate whether PBM could promote regeneration in sponges, we analysed Msi-1, which is considered to be an archaeocyte cell marker [44]. As shown in Figure 5A, at 24 h postexcision, Msi-1 was upregulated 5.19 ± 1.74-fold in the PBM samples compared with the control samples. At 120 h postexcision, Msi-1 did not differ between the groups. Moreover, in both the 120 h control and PMB samples, Msi-1 decreased significantly (0.3 ± 0.14 and 0.1 ± 0.007-fold, respectively) compared to the 24 h control samples, attesting to the transition from archaeocytes to exopinacocytes.

In the damaged tissue of higher animals, the endogenous wingless protein (Wnt) signal activates stem cells and contributes to repairs [45]. As shown in Figure 5B, at 24 h postexcision, a Wnt-like mRNA increased 1.7 ± 0.25-fold in the PBM samples compared with the control samples. There were no significant changes in Wnt gene expression between the groups at 120 h postexcision, but a significant increase in gene expression (1.61 ± 0.27-fold) was observed in the 120 h control samples compared to the 24 h controls.

At 24 h postexcision, fibrillar collagen had increased markedly by 49.35 ± 16.36-fold in the PBM samples compared with control samples (Figure 5C). At 120 h postexcision, there were no differences between the groups. In control samples, fibrillar collagen expression increased 2.35 ± 0.73-fold from 24 to 120 h postexcision. The opposite trend occurred in the PBM samples.

PBM did not affect the expression of the fibroblast growth factor (FGF) homologue at either of the analysed time points (Figure 5D).

Finally, we examined several transforming growth factor (TGF) family members. At 24 h postexcision, TGF3 was 0.68 ± 0.1-fold lower in the PBM samples compared with the control samples. There was no significant difference between the groups at 120 h postexcision, but a decrease in TGF3 gene expression in the 120 h control samples was observed compared to the 24 h controls (0.46± 0.18-fold), indicating a progression of the regeneration process (Figure 5E). TGF4 was not significantly different between the groups at either time point. However, in the control samples, TGF4 increased 2.38 ± 0.51-fold from 24 to 120 h postexcision (Figure 5F). At 24 h postexcision, there was a slight but significant reduction in TGF5 expression in the PBM samples compared with the controls (0.73 ± 0.14-fold lower). This reduction continued over time, and by 120 h postexcision, TGF5 gene expression was strongly downregulated by 0.46 ± 0.07-fold in the PBM samples compared with the 24 h control samples, resulting in significant downregulation in the 120 h control samples (Figure 5G). Finally, in the PBM samples, TGF6 expression was significantly upregulated by 1.36 ± 0.11-fold at 24 h postexcision compared with the control samples. There were no significant differences between the groups at 120 h postexcision. At that time, TGF6 was markedly downregulated in the PBM samples (0.4 ± 0.34-fold) when compared to the 24 h control samples (0.33 ± 0.19-fold) (Figure 5H).

### 2.4. Sample Temperature Analysis

Thermal camera measurements of the 810 nm-1 W PBM specimen surface registered 14.38 ± 0.21 °C before irradiation and 16.57 ± 0.33 °C after irradiation.

### 2.5. Oxidative Stress Evaluation

The 810 nm-1 W PBM could increase reactive oxygen species (ROS) generation through mitochondria respiratory chain-modulated activity [25]; hence, we evaluated the activity of oxidative stress enzymes, such as catalase and glutathione transferase (GST), and the oxidative marker malondialdehyde content in the regenerating *C. reniformis* samples 6 h after irradiation. As shown in Table 1, there were no significant differences in these markers between the groups, indicating that PBM does not generate oxidative stress in sponge tissues.

### 2.6. Prokaryotic Symbiont Quantification

Because PBM interacts with prokaryotic cells [46], we evaluated the microbial content in the regenerating edge of the *C. reniformis* samples. We isolated DNA from the samples 24 and 120 h after tissue injury and then calculated the prokaryotic DNA/sponge DNA ratio. As shown in Figure 6, at 24 h postexcision, the prokaryotic DNA slightly but significantly increased in the PBM samples compared with the control samples (1.19 ± 0.11-fold). At 120 h postexcision, there was no significant difference between the groups. All samples showed a substantial increase in prokaryotic DNA relative to the 24 h time point (control: 2.89 ± 0.81-fold; PBM: 2.52 ± 0.68-fold).

## 3. Discussion

The sponges are the earliest branching phylum of the animal kingdom [26]. Because they are a sister group to all other animal taxa and they have a very simple body plan, these members of Porifera have often been designated as ‘living fossils’ and are considered to be early animal ancestors [47]. However, some authors point out that although Parazoa/Eumetazoa branching has been dated back to millions of years ago, the Porifera phylum has been evolving for as long as any other modern animal lineage [48]. Hence, the great morphological simplicity of the sponges is combined with high molecular complexity. During the last 20 years, researchers have identified various molecular pathways in sponges that are typical of higher animals, such as innate immune response factors [34,49] and development and differentiation effectors [31,50,51]. Given these findings, we evaluated the effect of 810 nm-1 W PBM on sponge tissue regeneration to evaluate whether the molecular response is shared and has been maintained throughout evolution across the phyla of the animal kingdom.

Based on our qualitative analysis of the level of wound healing after 120 h of cutting, the irradiated samples showed a more rounded shape and smaller cutting edge than the controls. Likewise, the histological analysis of the regeneration edge at 24 h postinjury supports this observation. We confirmed the increased regeneration induced by 810 nm-1 W PBM in sponge tissue at the molecular level based on the strong upregulation of the fibrillar collagen gene 24 h post-injury. The extensive and fast deposition of new extracellular matrix components, such as fibrillar collagen, is essential for the tissue regeneration of *C. reniformis*, a marine sponge mainly formed by this fibrous protein. Moreover, the early, significant upregulation of the Msi-1 gene–considered a sponge stem cell marker [32,33,52]–indicated improved tissue regeneration induced by 810 nm-1 W PBM. The high number of Msi-1-positive cells in the regenerating edge of the PBM samples was further supported by immunostaining analysis.

For *C. reniformis*, we previously described a set of TGF family members–TGF3 and TGF6–whose expression profiles could have prostemness activity. Indeed, their result up-regulated in the early regeneration step and decreased during sponge tissue restoration. Additionally, TGF4 and TGF5 are considered to be prodifferentiating factors, as their expression increases only during the last phase of sponge tissue regeneration [31]. In the present study, we found a significant TGF6 upregulation in the PBM samples compared with untreated control samples during the early phase of regeneration, which supports the positive effect of 810 nm-1 W PBM in sponge tissue recovery. However, in irradiated samples, we also observed the downregulation of TGF3 (in the early phase) and TGF5 (120 h postinjury). These last two results could be due to the faster return to baseline expression levels of these genes in the PBM samples. As observed in the HSP60 and HSP70, PBM could act on TGF3 and TGF5 by stimulating a more rapid response in gene expression. Conversely, the expression of a sponge FGF homologue, as well as TGF4, seem to be not affected by irradiation, indicating that multiple, independent pathways regulate these genes.

PBM can induce a selective photothermal effect with local overheating occurring in pigmented tissue [53]. Although the sponge samples we used are pigmented, we did not observe a macroscopic temperature increase in the irradiated tissue and the medium. This finding is not unexpected; pigment deposition is prevalently localised to the external wall of the animal, but we irradiated the cut surface, which shows a quite uniform white appearance (Figure 1 and Figure 7A). The 2 °C temperature increase meets our previous data on *Paramecium* [9,11] irradiated through the same parameter used on the sponge. The low increment in temperature did not affect the cell metabolism and reproduction of these unicellular organisms. Additionally, the sponge surface temperature was immediately recovered in the aquarium. Therefore, we interpret the marked HSP60 and HSP70 gene upregulation 24 h postinjury as mainly due to the protein folding activity associated with the HSP family [54,55] rather than thermal stress. HSPs play a key role in the healing process. For example, in tendons, HSPs modulate the inflammatory response and reduce apoptosis by promoting the release of inflammatory cytokines and chemokines [54]. PBM treatment for regenerating tendon tissues induces HSP70 upregulation, improving wound healing [56]. Upregulation of the TNF and antiapoptotic Bcl-2 genes in our experimental model suggests that HSP70 and HSP60 could play the same role in response to wound healing across the animal kingdom and that they are the target of PBM even in lower metazoans. Overall, these data indicate that PBM-induced inflammation is not detrimental because it is concomitantly counteracted via the support of antiapoptotic membrane instability pathways. The disconnection between TNF signalling and apoptosis observed in this study is consistent with our previous work, where we exposed *C. reniformis* to quartz dust. This phenomenon could be explained by the structural differences in the intracellular region of the TNF receptor between sponges and higher animals [34].

Calcium-dependent nitric oxide synthesis in sponges, [42] mainly via NOS expressed in pinacocytes, is involved in the filter-feeding process by modulating whole-body contractions and reducing the incurrent system [57,58]. Therefore, the overall increase in NOS gene expression during sponge tissue regeneration observed in this study is not surprising because it is presumably associated with a progressive increase in pinacocytes around the regenerating edge, with evident sponge body contraction. Notably, there were no significant differences in NOS expression between the PBM and control samples, indicating that NOS expression is not affected by PBM through the 810 nm-1 W parameter in the sponges. However, these data do not allow us to exclude a local increase in NO release due to an increase in enzyme activity induced by 810 nm-1 W PBM, as previously observed in experimental models [10].

PBM treatment is extensively used in dentistry to reduce the bacterial load in the dental regenerating area [46]. Porifera are filter-feeding organisms that contain several symbiont microbial communities. Bacterial population and sponge tissue are considered a holobiont [59], in which the species-specific microbial group of marine sponges are thought to play a crucial role in animal metabolism and, ultimately, in their evolutionary success [60]. In our experimental model, at 120 h postinjury, there was an increase in the prokaryotic population along the regenerating edge of the control and PBM samples compared with the 24 h control samples. The downregulation of the TNF gene in the same samples leads us to exclude an infectious state; rather, these prokaryotes represent a normal symbiotic population that is recolonising the regenerating area. However, there was a small but significant increase in prokaryotes in the regenerating sponge tissue 24 h postinjury in the PBM samples. This seems to be at odds with what happens in the regenerating systems of higher organisms. The bactericidal effect of PBM is mainly related to pigmented prokaryotes, which experience local heating when exposed to light [46]. Conversely, the PBM effect on nonpigmented bacteria and particularly aerobic bacteria may be similar to what occurs in mitochondria: the 810 nm-1 W PBM stimulates the energetic cell metabolism of eukaryotic cells, from protozoa to mammals [6,9,24,46].

Our present and past data on the effect of PBM on life forms have highlighted that regeneration is a property of living matter, and regardless of its level of complexity, regeneration can be affected through 810 nm light. From this point of view, the *C. reniform* is data can be compared with the extensive information collected in our past work on the effect of 810 nm-1 W PBM on heterogeneous organisms from protozoa to humans [10,11,12,14,15,16,17,18,20,21,22,23,24,25,61].

Indeed, in the protozoan *P. primaurelia*, 810 nm-1 W PBM irradiation immediately affected oxygen consumption and adenosine triphosphate (ATP) production, which increased by more than 90% relative to the control, [11] and both endothelial and tumour human cells experienced similar effects [24,25]. Additionally, our work with mitochondria extracted from bovine liver showed that 810 nm-1 W PBM selectively interacts with complex III and IV of the mitochondria respiratory chain to increase activity while maintaining the ATP/O_2_ balance (P/O≈2.5); they may represent the PBM primary target. An energy boost from 810 nm-1 W PBM could explain and support the secondary events observed in our sponge specimens.

Data from *C. reniformis* and murine pre-osteoblasts and stem cells indicate that the conserved canonical Wnt/β-catenin and TGF signalling pathways are first activated by 810 nm-1 W PBM [19,20,62]. Moreover, Wnt signalling, as well as the Bcl-2/Bax ratio, which interacts to coordinate cell fate, are probably the next molecular targets. When moving from Porifera to Annelida [15,16] up to vertebrate animals [19,20] and humans [22,23], our data highlight the key role of PBM in modulating inflammatory processes to support faster recovery. The consistent action–effect mode induced by 810 nm-1 W PBM suggests a continuum of mechanisms, as demonstrated by the functional conservation of key molecular events regulating cell behaviour and the regeneration process of both related and unrelated taxa across wide phylogenetic distances. Additionally, regeneration induced by 810 nm-1 W PBM occurs in organisms where the process follows epimorphosis, morphallaxis, or a combination of the two; the regeneration ability has multiple hierarchical levels and complexities, and natural light (sunlight) interaction may not occur for the entire life of a cell. Therefore, the consistent effect of 810 nm-1 W PBM on all the irradiated organisms lies in the transfer of its photoenergy to the conserved photoacceptors included in all animal cells, which play a key role in the downstream pathway essential for the conserved Wnt/β-catenin and TGF signalling pathways.

Overall, the main processes observed in the regenerating tissue of higher organisms subjected to PBM–including a reduction in apoptosis, the synthesis of collagen, and an increase in stem cells–also occur in Porifera. These findings indicate that some of the molecular mechanisms activated in response to light are the result of an innate rather than adaptive competence in animal cells. This ability was acquired very early in evolution and has been conserved across the phyla of the animal kingdom.

## 4. Materials and Methods

### 4.1. Chemicals

All reagents were acquired from Sigma-Aldrich (Milan, Italy) unless otherwise stated.

### 4.2. Experimental Animals

*C. reniformis* (Nardo, 1847) specimens were collected at the Mount Portofino ‘Marine Natural Regional Park’ (Liguria, Italy) at depths of 10–20 m. During sampling and transport, the temperature was maintained at 14–15 °C. Short-term stabilisation was performed, as described by Pozzolini et al. [34,40]. Briefly, the sponges were stored at 14 °C in 200 L aquaria containing natural seawater collected in the same Portofino area with a salinity of 37‰ and equipped with an aeration system.

### 4.3. Experimental Design and Setting

The experimental conditions are shown in Figure 7. Three specimens of *C. reniformis* with a diameter of 6–8 cm were cut with a sterile scalpel into four equal parts (Figure 7A); 12 fragments were derived from it, namely a1, a2, a3, and a4; b1, b2, b3, and b4, and c1, c2, c3, and c4. Two fragments for each specimen (six fragments in total) were designated for the PBM samples. They were irradiated with an 810-nm diode laser (GaAlAs) device (Garda Laser, 7024 Negrar, Verona, Italy) equipped with our novel FT-HP, which is able to irradiate a spot area with a consistent energy distribution independently of the distance [61]. According to our previous studies, [9,11,12,13,14,16,17,18,19,20,21,22,24,25] we consistently administered 810nm-1W PBM therapy at 1 W of power in CW mode for 60 s on a circular spot of 1 cm^2^. The parameters allowed us to generate a power density of 1 W/cm^2^ and a fluence of 60 J/cm^2^ (energy administered = 60 J) (Figure 8). The other two fragments for each specimen (six fragments in total) were designated the control (CTR) samples. They were exposed to the same 810-nm diode laser and FT-HP, but the device was set to irradiate 0 W and 0 J for 60 s (810 nm-0 W PBM). A 635 nm red-light pointer (negligible power, <0.5 mW) was used in both treatments to visualise the exposed area and keep the experiment blinded [61]. After 60 s of treatment, all the fragments were transferred to the aquaria.

As shown in Figure 7B, during irradiation, the fragments were temporarily posed on a Petri dish; the FT-HP was fixed to a stand 0.5 cm away from the sample, and all were treated daily for a total of five irradiations at 0, 24, 48, 72 and 96 h.

The accuracy of the irradiated laser parameters was ensured with a Pronto-250 power meter (Gentec Electro-Optics, Inc. G2E Quebec City, Canada) [25,61]. To avoid beam reflection, the Petri dishes with the specimens were placed on an absorbent mat [25]. Adverse events due to a possible undesirable thermal effect were avoided by monitoring the irradiation with a FLIR ONE Pro-iOS thermal camera (dynamic range: −20 °C/+400 °C; resolution 0.1 °C; FLIR Systems, Inc. designs, Portland, OR, USA) [25,61]. Twenty-four hours after the first irradiation, three regenerating fragments for each experimental condition (three controls and three 810 nm-1 W PBM specimens) were recovered, and their regeneration edge was excised and processed for RNA and genomic prokaryotic DNA and histology. Lastly, the remaining fragments (three controls and three 810 nm-1 W PBM specimens) were collected 120 h postinjury and were processed for RNA and genomic prokaryotic DNA.

The same experimental setup was also used to evaluate possible oxidative stress induced by PBM (Figure 7C). The regeneration edge was excised 6 h after the first irradiation and processed for protein extraction and enzymatic assays to evaluate oxidative stress markers.

### 4.4. Histology and Immunofluorescence

After 24 h, both the control and PBM fragments were embedded in paraffin, and the regenerating edges were cut transversally into 5 µm sections used for histology. Sponge tissue organisation was observed after Masson’s trichrome staining. The presence of immunoreactivity for Msi-1, a marker of archaeocyte in sponge tissues, was evaluated by indirect immunofluorescence. Sections were incubated with polyclonal rabbit anti-Msi-1 antiserum (E-AB-17518-60; Elabscience, Houston, TX, USA), previously used on *C. reniformis* tissues [31], and then with the secondary anti-rabbit Alexa Fluor 594 (Abcam, Cambridge, UK). Nuclei were counterstained with DAPI. Negative controls were performed by omitting the primary antiserum. Histological slides were observed through a Leica DMRB light and epifluorescence microscope (Leica Microsystems, Wetzlar, Germany) equipped with Moticam 3+ (Motic Europe, Barcelona, Spain). A semiquantitative analysis of Msi-1 immunoreactivity on histological slides of the three specimens was performed by acquiring eight random photographs along the regenerating edges from two different slides for both the control and PBM sponges. The percentage of red fluorescent pixels on tissue-occupied pixels in each photograph was evaluated by using ImageJ [63].

### 4.5. Gene Expression Evaluation

After the set time points, as explained in Figure 7, the regenerating edges of the sponge portions were cut with a sterile scalpel, obtaining a thin layer of tissue about 3 mm thick that was rapidly chopped into smaller pieces and put into an adequate volume of Isol-RNA Lysis Reagent (5 Prime GmbH, Hilden, Germany) for total RNA extraction. The poly-A fraction was then isolated with the FastTrack^®^ MAG mRNA isolation kit (Life Technologies, Milan, Italy) following the manufacturer’s instructions. Complementary DNA (cDNA) was synthesised by Revert Aid Reverse Transcriptase (Thermo Fisher Scientific, Milan, Italy), starting from 1 μg of purified RNA from each sample. Each PCR reaction was carried out in a final volume of 15 μL containing: 1× iQ SYBR^®^Green master mix (Bio-Rad, Milan, Italy), 0.2 μM of each primer, and 3 μL of a 1:5 dilution of cDNA. For each sample, the analysis was conducted in triplicate. The following thermal profile was used: initial denaturation at 95 °C for 3 min, followed by 45 cycles with denaturation at 95 °C for 15 s, and annealing and elongation at 60 °C for 60 s. Fluorescence was measured at the end of each elongation step. The values were normalised to GAPDH expression (reference gene). All the PCR primers (see Appendix A) were designed using Beacon Designer 7.0 software (Premier Biosoft International, Palo Alto, CA, USA) and were provided by TibMolBiol (Genova, Italy). Data analyses were performed with the DNA Engine Opticon^®^ 3 Real-Time Detection System Software program (3.03 version). To calculate the relative gene expression compared to an untreated (control) calibrator sample, the comparative threshold Ct method was used for gene expression analysis in the iCycler iQ Real-Time Detection System Software^®^ (2004 Bio-Rad, Hercules, CA, USA). The data are presented as the mean ± standard deviation of three independent experiments performed in triplicate.

Statistical analyses were performed using a one-way analysis of variance followed by Tukey’s posthoc test (GraphPad Software, Inc., San Diego, CA, USA). A *p*-value < 0.05 was considered significant.

### 4.6. Oxidative Stress Enzyme Assays

To quantify stress-related enzymatic activity, a 2–3 mm thick slice of tissue from the regenerating edge of each sample was cut off with a sterile scalpel (Figure 7C). The tissue slice was then manually chopped into small pieces that were put into cold and sterile phosphate-buffered saline (PBS), pH 7.5. A bead mill homogenisation with the Tissue-Lyser system was performed (two cycles of 1 min each). The suspensions were centrifuged (18,000× *g* at 4 °C for 20 min) to remove any insoluble molecules; the supernatant was stored at −80 °C for further experiments. The extracted protein concentration was determined with a Bradford assay.

#### 4.6.1. Catalase Activity

Catalase activity was evaluated by measuring the decomposition of hydrogen peroxide (H_2_O_2_) following Claiborne’s method [64]. Each reaction mixture contained 890 μL of 100 mM PBS (pH 7.5), 100 μL of 500 mM H_2_O_2_, and 10 μL of the sample, added right before testing. Catalase activity in the sample was evaluated by measuring the absorbance of each sample at 240 nm every 15 s for 15 min, using a Beckman spectrophotometer (DU 640, Indianapolis, IN, USA). The results are expressed as μmol H_2_O_2_/min/mg proteins.

#### 4.6.2. GST Activity

GST activity was evaluated following the method described in Friedberg et al. [65] using 1-chloro-2,4-dinitrobenzene (CDNB) and the reduced glutathione form (GSH) as substrates. CDNB was dissolved in 100% ethanol, and GSH was dissolved in 100 mM Tris-HCl (pH 7.4). The assay mixture contained 1 mL of 100 mM PBS (pH 7.5), 50 of 1 mM CDNB, 50 μL of 5 mM GSH, and 50 μL of sample, added right before measurement. GST activity was evaluated by measuring the absorbance at 340 nm every 15 s for 5 min using a Beckman spectrophotometer (DU 640). The results are presented as nmol GSH-CDNB complex/min/mg proteins.

#### 4.6.3. Malondialdehyde (MDA) Content Determination

MDA, a thiobarbituric acid reactive species (TBARS) resulting from lipid peroxidation processes, was quantified in the samples following the method of Buege and Aust [66] with a few modifications. For each sample, an assay mixture was prepared, containing 1 mL of 0.67% thiobarbituric acid, 400 μL of 20% trichloroacetic acid, and 100 μL of the sample. Solutions were incubated at 100 °C for 15 min, then centrifuged at 4000× *g* at 4 °C for 10 min. The supernatant was kept, the rest discarded, and finally read at 512 nm using a Beckman spectrophotometer (DU 640). The MDA content is expressed as nmol TBARS/mg proteins.

### 4.7. Symbiont Bacteria Quantification

Genomic DNA was extracted from the slices of tissue of the *C. reniformis* specimens stimulated as described in Section 4.3., with the CTAB method. [67] The relative number of bacterial genes in each sample was quantified by real-time PCR analysis using a Chromo4 instrument (MJ Research, Bio-Rad, Hercules, CA, USA), with the *C. reniformis* GAPDH used as the reference gene as described previously by Pozzolini et al. [37].

## 5. Conclusions

In conclusion, our results comply with both the primary variable and endpoint. Indeed, PBM administered at 810 nm, 1 W, and 60 J and applied for 60 s with an FT-HP on a circular area of 1 cm^2^ affects *C. reniformis* regeneration, which appears to occur faster than in the control. The effect is consistent with our previous data on a wide range of animals and humans, showing cell pathways conserved through the evolution of life forms, which appear to be influenced by NIR light.

Concerning future perspectives, the increment of collagen deposition induced by PBM treatment observed in our study could open its usage to sponge mariculture techniques for biotechnological purposes. Lastly, although the 810 nm wavelength of the sun does not usually reach *C. reniformis* in nature, the evidence of interactions between light and sponge tissue suggests a deeper investigation of the influence of sunlight wavelengths on sponge metabolism, growth, and recovery.

## Figures and Tables

**Figure 1 ijms-24-00226-f001:**
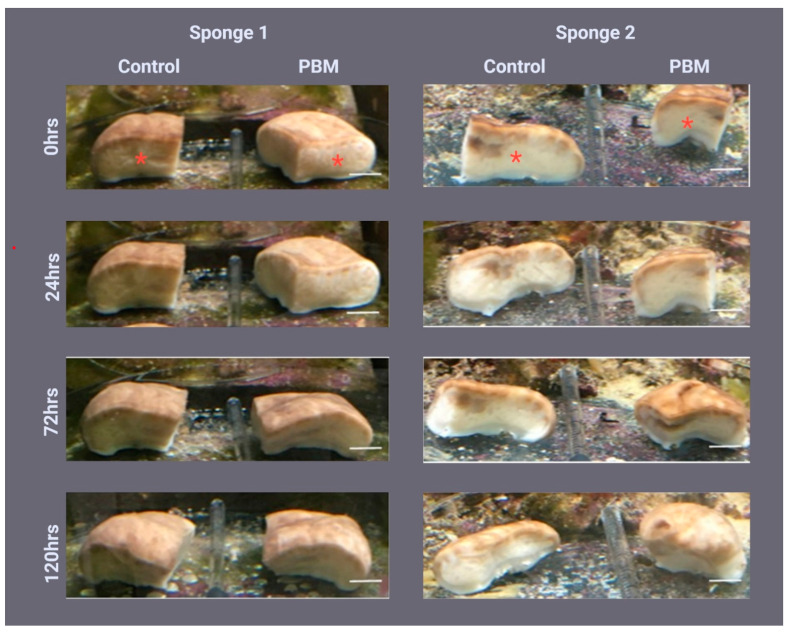
Effect of photo-biomodulation (PBM) on *Chondrosia reniformis* morphology during tissue regeneration. The regeneration profile of two *C. reniformis* specimens immediately (0 h) and 24, 72, and 120 h after being cut into four parts, daily irradiated on the regenerating edge with 810 nm-1 W PBM. PBM = samples irradiated with 810 nm-1 W PBM; control = samples irradiated with 810 nm-0 W PBM. Scale bars are 1 cm. The centre of the irradiated area is indicated by the red asterisk.

**Figure 2 ijms-24-00226-f002:**
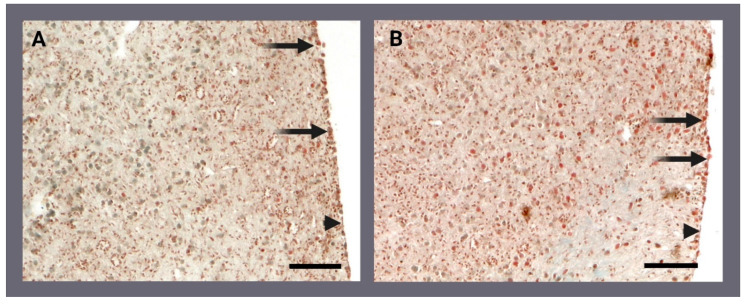
Histological sections showing the regenerating edges of *Chondrosia reniformis* fragments 24 h after being cut. The (**A**) control (810 nm-0 W photo-biomodulation [PBM]) and (**B**) irradiated (810 nm-1 W PBM) fragments were stained with Masson’s trichrome. Some spherulous cells (arrows) and exopinacocyte-like cells (arrowheads) are visible on the regenerating edge. The scale bars are 50 µm.

**Figure 3 ijms-24-00226-f003:**
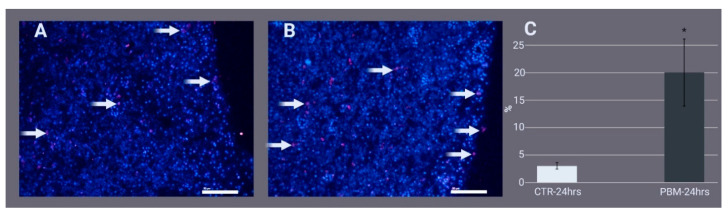
Histological sections showing the regenerating edges of *Chondrosia reniformis* fragments 24 h after being cut. Immunofluorescence of the (**A**) control (810 nm-0 W photo-biomodulation [PBM]) and (**B**) irradiated (810 nm-1 W PBM) fragments stained for Msi-1 (red) and DAPI (blue). Archaeocytes are scattered in the sponge tissue (arrows). The scale bars are 50 µm. (**C**) Semiquantitative Msi-1 immunofluorescence analysis: the percentage of red fluorescence pixels on total tissue-occupied pixels. Student’s *t*-test, *p* < 0.001 (*). CTR = control. Each bar represents the mean ± standard deviation of three independent experiments performed in triplicate. The negative control is shown in Appendix A.

**Figure 4 ijms-24-00226-f004:**
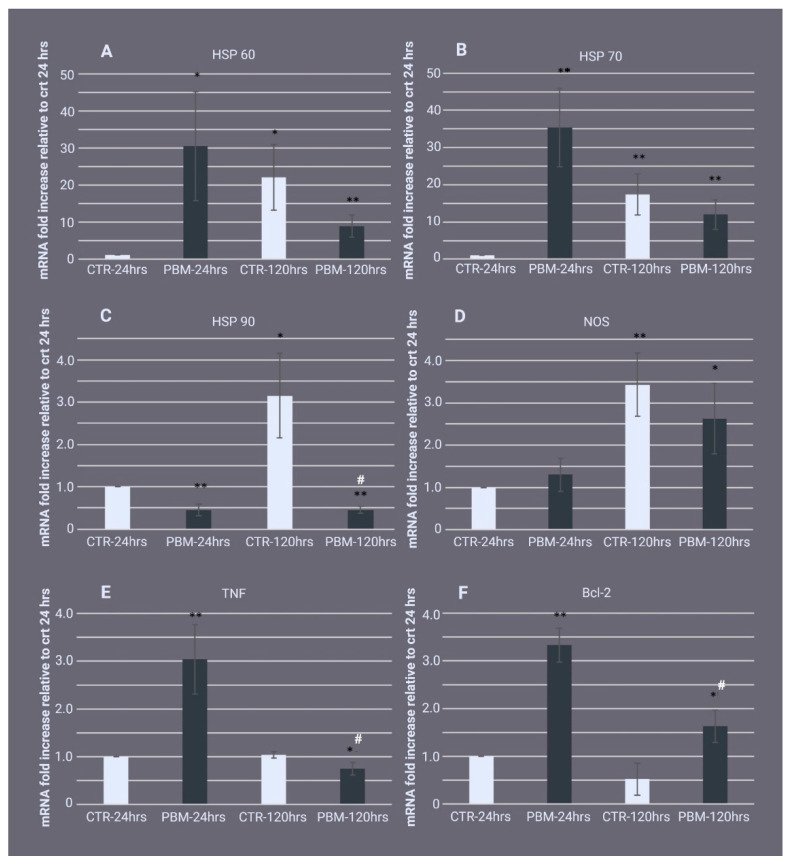
The effect of photo-biomodulation (PBM) on the expression of genes related to thermal stress, inflammation, and apoptosis. The real-time PCR gene expression analysis of (**A**) heat shock protein 60 (HSP60), (**B**) HSP70, (**C**) HSP90, (**D**) nitric oxide synthase (NOS), (**E**) tumour necrosis factor (TNF), and (**F**) Bcl-2 (**F**) in regenerating *Chondrosia reniformis* tissue 24 h (PBM-24 h) or 120 h (PBM-120 h) after PBM. The control groups (CTR-24 h and CTR-120 h) are the respective 810 nm-0 W PBM nonirradiated samples (0 J). The data are expressed as the messenger RNA (mRNA) fold-increase relative to the CTR-24 h sample and are normalised to GAPDH expression. Each bar represents the mean ± standard deviation of the three independent experiments performed in triplicate. The data were analysed with one-way analysis of variance (ANOVA) followed by Tukey’s test (ANOVA *p*-values: (**A**) *p* = 0.013071; (**B**) *p* = 0.001023; (**C**) *p* = 0.000511; (**D**) *p* = 0.003392; (**E**) *p* = 0.000192; (**F**) *p* = 0.000014). The asterisks indicate significant differences between groups based on Tukey’s test versus CTR-24hrs; * *p* < 0.05; ** *p* < 0.01. The hash symbol indicates significant differences between groups based on Tukey’s test versus CTR-120 hrs, # *p* < 0.05.

**Figure 5 ijms-24-00226-f005:**
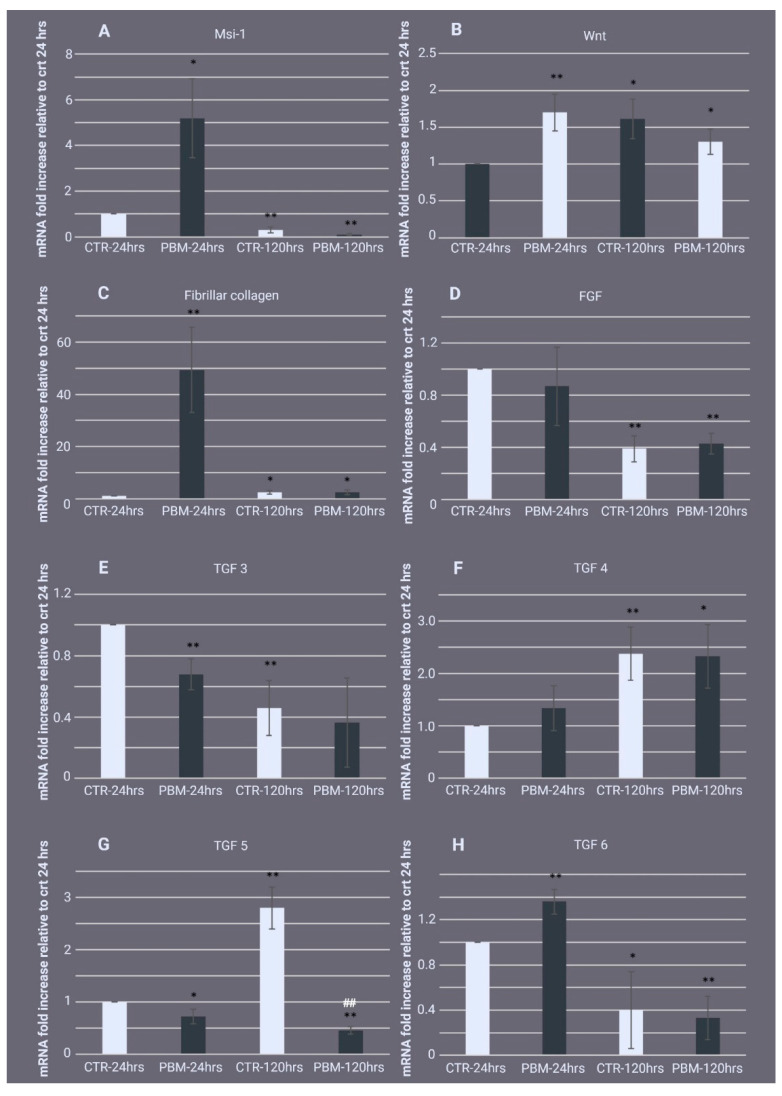
Effect of photo-biomodulation (PBM) on the expression of proliferation/differentiation, extracellular matrix synthesis, and growth factor genes. The real-time PCR gene expression quantification of (**A**) the pluripotent stem cell-inducer gene Musashi 1 (Msi-1); (**B**) the endogenous wingless protein (Wnt) signalling pathway; (**C**) fibrillar collagen; (**D**) fibroblast growth factor (FGF), and (**E**) transforming growth factor 3 (TGF3), (**F**) TGF4, (**G**) TGF5, and (**H**) TGF6 (**H**) in regenerating *Chondrosia reniformis* tissue 24 h (PBM-24 h) and 120 h (PBM-120 h) after PBM. The control groups (CTR-24 h and CTR-120 h) are the respective 810 nm-0 W PBM nonirradiated samples (0 J). The data are expressed as the messenger RNA (mRNA)-fold increase relative to CTR-24 h and are normalised to GAPDH expression. Each bar represents the mean ± standard deviation of three independent experiments performed in triplicate. The data were analysed with one-way analysis of variance (ANOVA) followed by Tukey’s test (ANOVA *p*-values: (**A**), *p* = 0.013071; (**B**), *p* = 0.001023; (**C**), *p* = 0.000511; (**D**), *p* = 0.003392; (**E**), *p* = 0.000192; (**F**), *p* = 0.000014). The asterisks indicate significant differences between the groups based on Tukey’s test versus CTR-24 h, * *p* < 0.05, ** *p* < 0.01. The hash symbol indicates significant differences between groups based on Tukey’s test versus CTR-120 h, ## *p* < 0.01.

**Figure 6 ijms-24-00226-f006:**
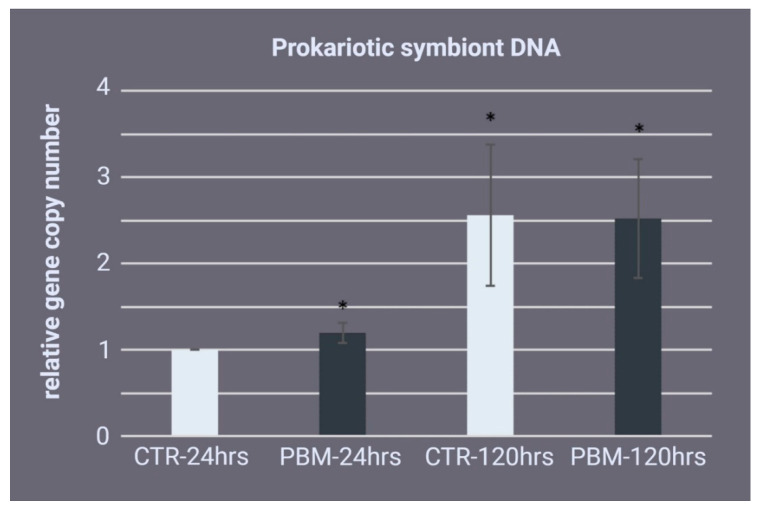
Prokaryotic symbiont DNA quantification. Analysis of the sponge prokaryotic symbiont population based on real-time PCR DNA quantification in the regenerating *Chondrosia reniformis* specimens 24 h postinjury and the first dose of 810 nm-1 W photo-biomodulation (PBM-24 h) and 120 h postinjury (PBM-120 h). The control groups (CTR-24 h and CTR-120 h) are the respective non-irradiated samples. The data are expressed as the prokaryotic DNA gene copy number relative to CTR-24 h and are normalised to the GAPDH gene copy number. Each bar represents the mean ± standard deviation of the three independent experiments performed in triplicate. The data were analysed with a one-way analysis of variance (ANOVA) followed by Tukey’s test (ANOVA *p* = 0.006047). The asterisks indicate a significant difference between groups based on Tukey’s test (versus CTR-24 h, * *p* < 0.05).

**Figure 7 ijms-24-00226-f007:**
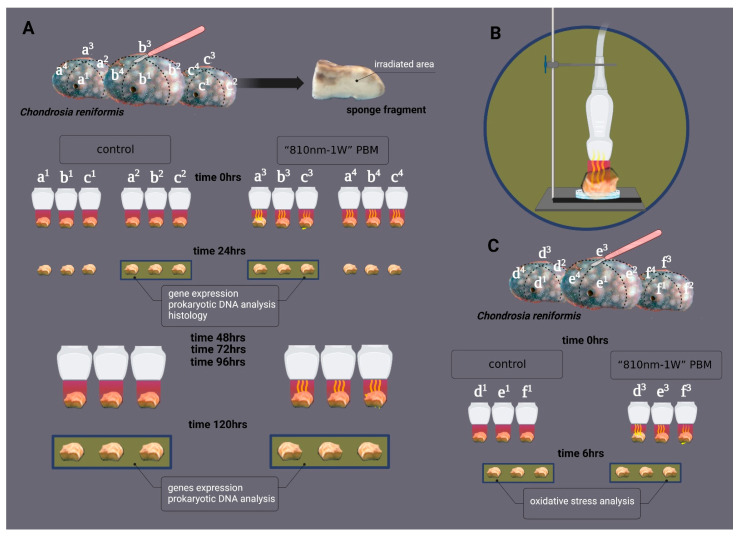
Summary of the experimental design used in the study. (**A**) Three *Chondrosia reniformis* were cut into four fragments (a1, a2, a3, and a4; b1, b2, b3, and b4; c1, c2, c3, and c4). A total of 12 fragments were obtained and were equally subdivided into two groups. (**B**) Six fragments (a1, a2; b1, b2; c1, c2) were irradiated with 810 nm-0 W, 0 J (control), and six fragments (a3, a4; b3, b4; c3, c4) were exposed to 810 nm-1 W PBM. Irradiation was performed daily for five days, and the specimens were analysed for gene expression, prokaryotic DNA analysis, and histology. (**C**) Three *C. reniformis* were cut into four fragments (d1, d2, d3, and d4; e1, e2, e3, and e4; f1, f2, f3, and f4). A total of 12 fragments were obtained, and six of these were equally subdivided into two groups. Three fragments (d1, d2; e1, e2; f1, f2) were irradiated with 810 nm-0 W, 0 J (control), and three fragments (d3, d4; e3, e4; f3, f4) were exposed to 810 nm-1 W PBM. Irradiation was performed on the first day (0 h), and the specimens were collected after 6 h for oxidative stress analysis. Irradiations were performed with a flat-top hand-piece fixed to a stand 0.5 cm from the sample (**B**).

**Figure 8 ijms-24-00226-f008:**
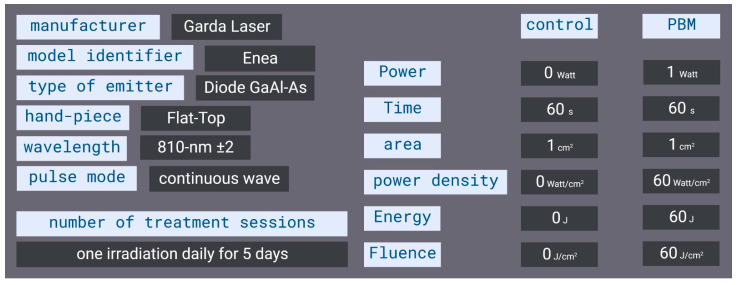
Outline of the PBM treatment parameters used.

**Table 1 ijms-24-00226-t001:** Oxidative stress enzyme activity and malondialdehyde content in *Chondrosia reniformis* 6 h postirradiation. The samples were normalized to total protein content.

Sample	Catalase Activity (μmol H_2_O_2_/min/mg Protein)	GST Activity (μmol GST-CDNB/min/mg Protein)	Malondialdehyde Content (nmol/mg Protein)
CTR	6.03 ± 0.1	0.46 ± 0.19	172.8 ± 75.37
PBM	5.41 ± 0.14	0.52 ± 0.12	225.61 ± 88.01

The data are presented as the mean ± standard deviation. CTR = 810 nm-0 W nonirradiated specimens (control); GST = glutathione transferase; PBM = 810 nm-1 W irradiated specimens.

## Data Availability

Data are available on request from the authors.

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
