# Peer review of "Near-Infrared 810 nm Light Affects Porifera Chondrosia reniformis (Nardo, 1847) Regeneration: Molecular Implications and Evolutionary Considerations of Photobiomodulation–Animal Cell Interaction"

_ijms, 2022, doi:10.3390/ijms24010226_

Round 1
Reviewer 1 Report
Dear authors,
The study is very interesting, but it needs some changes in my point of view.
Abstract:
I missed the number of specimens, the results and the conclusion.
Introduction:
Do not write the material and method in the introduction, but the objective and justification of the study (last paragraph).
The text is in the wrong order. Material and method come before RESULTS.
Material and Methods
Start with item 4.3 and not with 4.1.
There is a need to organize to facilitate understanding.
Results:
Write in the text the statistics used, and not before the conclusion.
Improve the legend of figures, and place important information in the text of the results.
Conclusion
The conclusion needs to be clearer, it must respond to the objectives, which in my view are not clear.
Author Response
Dear Reviewer 1,
First of all, thank you so much for appreciating our paper and the precious advice that improved our work.
Please find in the journal’s template our revisions highlighted in yellow.
Point-by-point answers to your comments are listed below;
Q: Abstract: I missed the number of specimens, the results and the conclusion.
A: We agree that an extensive description of the work may be useful for the readers. However, the very limited number of words suggested by the author’s guidelines and the request of dividing the abstract with a background, method, results and conclusion, limited the description of many parts of the work. To meet your request, we changed some sentences, added the specimens’ numbers, and added the data on the TGF which supports the faster regeneration process observed in the irradiated fragments.
Q: Introduction: Do not write the material and method in the introduction, but the objective and justification of the study (last paragraph).
A: We modified the introduction’s last section and added the predictor variable and the primary endpoint of our work.
Q: The text is in the wrong order. Material and method come before RESULTS.
A: We used the Microsoft Word template and the order suggested by the Journal is: 1. Introduction; 2. Results; 3. Discussion; 4. Materials and Methods; 5. Conclusion.
Q: Results: Write in the text the statistics used, and not before the conclusion. Improve the legend of figures, and place important information in the text of the results.
A: The statistic used was written in the text as suggested. Some legends were improved in the results section.
Q: Conclusion: The conclusion needs to be clearer, it must respond to the objectives, which in my view are not clear.
A: We modified this section and clearly identified the conclusion, which complies with the primary both variable and endpoint added in the introduction, and the future perspectives in biotechnological and ecological fields.
Sincerely,
Andrea Amaroli

Reviewer 2 Report
The authors of the manuscript "810-nm near-infrared light affects Porifera Chondrosia reniformis Nardo, 1847 regeneration: molecular implications and evolutionary considerations of photobiomodulation–animal cell interaction" are reporting on the effect of NIR light treatment on the regeneration of the sponge Chrondrosia reniformis.
Although the idea the authors bare in mind is a good one the manuscript lacks novelty in the field, errors in reporting materials, and needs improvement in several areas. The whole manuscript would benefit from language editing and improving readability.
Minor comments:
1. Language and meaning of the abstract have to be improved.
2. Fig.2 Masson's-Trichrome staining - why did the authors opt for this staining and draw conclusions about certain cell types from that? I can imagine that there might be more cells or a certain cell type present but the staining doesn't identify what the authors propose. Secondly, to emphasize their claim authors should have quantified staining.
3. In Fig3. the number of samples is missing also I couldn't find any evidence that the Antibody is really specific to Msi-1 in Chondrosia - negative controls should be shown. Secondly, I couldn't find the antibody on the Invitrogen website. I found it to be from Elabscience. Also, the data generated from the IFs seems to be only from 2 different sponges - this is not enough.
4. Ref. 9 and Ref. 10 in the Bibliography are the same papers
5. The conclusion that is drawn from Table 1 that PBMT doesn't interfere with oxidative stress might be too far-fetched cause only one-time point was investigated.
Major comments:
1. The claims that the authors make from Fig.1 I cannot agree on due to the lack of resolution and quantification of such images. It also seems that the sponges were not cut into equal size pieces e.g. standardized in size. This is of concern, as differently sized wounds can heal differently and thus could impact the outcomes in regeneration.
2. Fig.4 gene expression data is missing explanations/interpretations in the main body section. The statistics shown in the graphs are also misleading as everything is referenced to the CTRL at 24h. Additionally, which Wnt gene is investigated as there exist several different Wnt genes (at least in better known species) and are known to execute different things in regards to stem cells and repair.
3. Supplemental table. The table gives the genes, primer sequence, and accession numbers. Many of the accession numbers don't exist - or I can't get any results when searching the internet. I couldn't find any match for all HSP genes, Wnt, Msi-1, and all TGFs. Where did the authors find those primers/accession numbers?
4. Fig. 6 I find it hard to understand how they measured the amount of prokaryotic DNA vs sponge DNA.
The manuscript needs numerous improvements throughout and especially a better interpretation of the results in that given section. Also a thorough explanation about the discrepancy with the accession numbers and the Msi-1 antibody should be provided.
Author Response
Dear Reviewer 2,
First of all, thank you so much for appreciating our paper and the precious advice that improved our work.
Please find in the journal’s template our revisions highlighted in green.
Point-by-point answers to your comments are listed below;
Q: The language and meaning of the abstract have to be improved
A: According to Reviewer 1 requests, we implemented and modified the abstract. We also corrected some errors. However, after acceptance, the Journal will provide a careful grammar English revision before publication.
Q: Fig.2 Masson's-Trichrome staining - why did the authors opt for this staining and draw conclusions about certain cell types from that? I can imagine that there might be more cells or a certain cell type present but the staining doesn't identify what the authors propose. Secondly, to emphasize their claim authors should have quantified staining.
A: Trichrome staining methods, such as Masson’s Trichrome, have been often chosen as routine staining in Porifera histology. See for example: Harrison and Cowden, 1974 doi: 10.1111/j.1432-0436.1975.tb01448.x; Rosell and Uriz, 1991 doi: 10.1080/00785326.1991.10429741; Bonasoro et al. 2001, 10.1007/PL00008497; Pozzolini et al., 2016 doi: 10.1086/BBLv230n3p220; Pozzolini et al. 2019 doi:10.1242/jeb.207894; Stocchino et al. 2021 doi: 10.1080/24750263.2020.1862316. The regenerative process at the edge of a cut in C. reniformis was described elsewhere (Pozzolini et al., 2019, doi:10.1242/jeb.207894), and spherulous cells and pinacocytes are already known as the cell types present on the regenerative surface. They are also quite distinguishable according to their shape (roundish or flatten) but, on the other hand, as cell identification is not so precise as in the immunostained sections, we were not confident in a cell count to demonstrate differences among the treatment.
Q: In Fig3. the number of samples is missing also I couldn't find any evidence that the Antibody is really specific to Msi-1 in Chondrosia - negative controls should be shown. Secondly, I couldn't find the antibody on the Invitrogen website. I found it to be from Elabscience. Also, the data generated from the IFs seems to be only from 2 different sponges - this is not enough.
A: This antibody was used before in C. reniformis (see Pozzolini et al., 2019 doi:10.1242/jeb.207894). The quantification of msi-1 gene expression, obtained with molecular techniques, has been performed for all the different regeneration stages and treatments considered. Instead, immunofluorescence is here intended as a qualitative, descriptive technique, and it was not performed for all the stages. Nevertheless, we attempted a fluorescence quantification to show that the fluorescence seems to go along with what is shown by the molecular analyses. A negative control will be added in the supplementary materials. The antibody was actually purchased from Elabscience and the Invitrogen indication was a mistake. Thank you.
Q: Ref. 9 and Ref. 10 in the Bibliography are the same papers
A: Sorry, we deleted Ref 10.
Q: The conclusion that is drawn from Table 1 that PBMT doesn't interfere with oxidative stress might be too far-fetched cause only one-time point was investigated.
A: PBM directly acts on mitochondrial targets involved in the respiratory chain. We previously showed that 810nm-1W PBM positively affects the mitochondria respiratory chain from protozoa to human cells and that the effect was correlated to the increment of ATP production without additive oxidative stress; (ln 325-333) Conversely, in tumour cells an increment of oxidative stress was described by our previous data. Additionally, PBM can have an antioxidative effect after injury. Therefore, in our study, we investigated the effect of irradiation a short time after laser administration to understand if the regenerative process was or was not influenced by ROS in the first phases of the process.
Q: The claims that the authors make from Fig.1 I cannot agree on due to the lack of resolution and quantification of such images. It also seems that the sponges were not cut into equal size pieces e.g. standardized in size. This is of concern, as differently sized wounds can heal differently and thus could impact the outcomes in regeneration.
A: The sponges were not cut into equal pieces because, to minimize the regeneration edge, it was decided to cut the entire body into four parts as similar as possible, without altering the aquifer system too much. Fig. 1 has the sole purpose of giving a qualitative indication and for this reason, there are no quantifications and statistical replies. However, we have removed the sentences t lines 257-260.
Q: Fig.4 gene expression data is missing explanations/interpretations in the main body section. The statistics shown in the graphs are also misleading as everything is referenced to the CTRL at 24h. Additionally, which Wnt gene is investigated as there exist several different Wnt genes (at least in better known species) and are known to execute different things in regards to stem cells and repair.
A: Some more details on the results of Figure 4 have been included in the text as also suggested by Reviewer 1
For some genes the significance was also indicated between the ctr.120 and PBM-120 pairs, however, to make the graphs clearer, we have reported the significance levels of several pairs with different symbols. Thank you
Regarding the transcript of the wnt gene analysed in the present study, it is the only transcript whose putative sequence provided with the BLASTp algorithm an acceptable E value with already described wnt sequences (1e-130 with QXY82380.1)
Q: Supplemental table. The table gives the genes, primer sequence, and accession numbers. Many of the accession numbers don't exist - or I can't get any results when searching the internet. I couldn't find any match for all HSP genes, Wnt, Msi-1, and all TGFs. Where did the authors find those primers/accession numbers?
A: The authors thank you for the comment, in relation to the sequences of wnt, nos, FGF, HSP70, HSP60 and HSP90 they have been deposited in genebank which has provided the accession numbers reported in the supplemental table however, we are sorry to see that the public release date has not yet been reached. For this reason, we provided the fasta sequences following the Table S1. Regarding the MsiI, we apologize, an incorrect accession number was entered, we have corrected it. All others accession numbers are correct and are found in NCBI's nucleotide database.
Q: Fig. 6 I find it hard to understand how they measured the amount of prokaryotic DNA vs sponge DNA.
A: The quantification approach used by the authors in the present study is relative and not absolute. The prokaryotic DNA level with respect to the sponge DNA is compared in the various samples. Total DNA (eukaryotic and prokaryotic) was isolated from each tissue sample and 100 mg was used to amplify 16S gene using 16S prokaryotic universal primers and the C. reniformis GAPDH gene. By exploiting the DDCt method was possible to normalize the 16 S copy numbers to sponge DNA (GAPDH gene). We have implemented the methodology description in the text.
Sincerely,
Andrea Amaroli

Round 2
Reviewer 2 Report
2nd Review report on PBM & C. Reniformis
Although the authors did address some of the concerns raised by the reviewer and improved the manuscript slightly, there are still several issues:
· Authors should indicate the site of irradiation for each image in Fig 1. It is not evident from the photographs shown. In addition, a quantification of the wound edge should be performed to demonstrate wound healing and the regenerative effect of PBM treatment.
· The authors still don’t state the biological replicates for Fig 3 in the figure legends. Even if the fluorescence analysis was there to provide additional support for the molecular analyses, a statistical test is only valid when at least 3 different samples have been used, an n=2 is not sufficient for a statistical test. So the authors either add a 3rd analysis or take out the statistics.
· Fig. 5F – this graph is not in line with the presentation of all the other graphs – where the 24h CTRL is set to 1. This needs to be corrected and the values probably re-calculated.
· The authors do not discuss the almost 2°C increase in temperature after PBM treatment and whether this could have any implications on their results.
· Regarding the mRNA data sets – I feel the interpretation of the data and the language used in the results section is sometimes misleading and not accurate of what the RNA data actually represents – I see the RNA data as follows:
o Hsp60 & Hsp70 -> PBMT just leads to a quicker response, potentially meaning that PBM activates WH or stress response faster, overall in a time course this is probably also why there is less mRNA at the PBM-120h samples
o Line 143 please correct 12h to 120h ctrl
o Hsp90 -> authors state that Hsp90 is decreased this might only be true for the 24h timepoints because if you compare PBM 24h and 120h there is no difference in gene expression and without additional time points in between it is pure speculation whether the expression decreased or not; secondly I see PBM here as a negative regulator of Hsp90
o TNF – should be rephrased to TNF levels probably dropped back to pre-PBM / baseline levels, as the difference between CTRL and PBM at this point is marginal
· DISCUSSION section
o The discussion of the TGF data contradicts the findings in the results section e.g.
§ TGF 6 is increased and not decreased; TGF5 expression is not increased in the later time points with PBM
The discussion section lacks an interpretation of the regenerative potential of PBM on C. reniformis and rather discussed the conservation across species
Author Response
Moscow, 16th December 2022
Dear Reviewer 2,
First of all, thank you so much for appreciating our paper and for the precious advice that improved our work.
Please find in the journal’s template our revisions highlighted in yellow.
Point-by-point answers to your comments are listed below;
Q1. Authors should indicate the site of irradiation for each image in Fig 1. It is not evident from the photographs shown. In addition, a quantification of the wound edge should be performed to demonstrate wound healing and the regenerative effect of PBM treatment.
A1. We added an asterisk in the centre of the irradiated area and described the asterisk’s meaning in the text. We measured the areas of the samples before and after irradiation and in the controls. The program ImageJ was employed to investigate surface areas. We included a sentence in the results and a figure in the supplementary material (figure S2).
Q2. The authors still don’t state the biological replicates for Fig 3 in the figure legends. Even if the fluorescence analysis was there to provide additional support for the molecular analyses, a statistical test is only valid when at least 3 different samples have been used, an n=2 is not sufficient for a statistical test. So the authors either add a 3rd analysis or take out the statistics.
A2. All the experiments were performed in triplicate. Therefore, “Two different slides for both control and PBM” has to be understood as two slides for each of the three samples. We added a short sentence to better clarify the misunderstanding.
Q3. Fig. 5F – this graph is not in line with the presentation of all the other graphs – where the 24h CTRL is set to 1. This needs to be corrected and the values probably re-calculated.
A3. Sorry for the typo. The graduated axis moved during the image generation. We adjusted the image and the figure.
Q4. The authors do not discuss the almost 2°C increase in temperature after PBM treatment and whether this could have any implications on their results.
A4. The increment was not discussed because it is a very low increase and only at the end of 60 seconds of irradiation. Plus, the specimens were restored in the aquarium immediately after irradiation and the temperature returned to the water temperature. In photobiomodulation, the range of increase observed in our study is not correlated to effects. To comply with your request we added a sentence to the discussion.
Q5. Regarding the mRNA data sets – I feel the interpretation of the data and the language used in the results section is sometimes misleading and not accurate of what the RNA data actually represents – I see the RNA data as follows:
o Hsp60 & Hsp70 -> PBMT just leads to a quicker response, potentially meaning that PBM activates WH or stress response faster, overall in a time course this is probably also why there is less mRNA at the PBM-120h samples
A5. We added some sentences in the results section.
Q6. o Line 143 please correct 12h to 120h ctrl
A6. We correct the typo.
Q7. o Hsp90 -> authors state that Hsp90 is decreased this might only be true for the 24h timepoints because if you compare PBM 24h and 120h there is no difference in gene expression and without additional time points in between it is pure speculation whether the expression decreased or not; secondly I see PBM here as a negative regulator of Hsp90
A7. We added some sentences in the results section.
Q8. o TNF – should be rephrased to TNF levels probably dropped back to pre-PBM / baseline levels, as the difference between CTRL and PBM at this point is marginal
A8. We added some sentences in the results section.
Q9. o The discussion of the TGF data contradicts the findings in the results section e.g.
- TGF 6 is increased and not decreased; TGF5 expression is not increased in the later time points with PBM
A9. We changed the text.
Q10. The discussion section lacks an interpretation of the regenerative potential of PBM on C. reniformis and rather discussed the conservation across species
A10. I do not agree with this conclusion. The entire article shows the effect of PBM on the sponge and the effect on many markers investigated is carefully interpreted. In accordance with the title of our paper, we also compared the results on the sponge with those previously obtained by our team on Paramecium, animals and human. This is the first time that a team performed the same PBM treatment on organisms with different levels of complexity. Therefore, the comparison is a strong section of the paper, not a weakness.
Hoping to encounter your appreciation,
Sincerely,
Andrea Amaroli
Adjunct Professor
Faculty of Dentistry
Department of Orthopedic Dentistry
First Moscow State Medical University (Sechenov University)

Round 3
